# *MUC5B* Polymorphism in Patients with Idiopathic Pulmonary Fibrosis—Does It Really Matter?

**DOI:** 10.3390/ijms26052218

**Published:** 2025-02-28

**Authors:** Katarzyna B. Lewandowska, Urszula Lechowicz, Adriana Roży, Maria Falis, Katarzyna Błasińska, Lilia Jakubowska, Monika Franczuk, Beata Żołnowska, Justyna Gryczka-Wróbel, Piotr Radwan-Rohrenschef, Anna Lewandowska, Olimpia Witczak-Jankowska, Małgorzata Sobiecka, Monika Szturmowicz, Witold Z. Tomkowski

**Affiliations:** 1First Department of Lung Diseases, National Research Institute of Tuberculosis and Lung Diseases, Płocka 26, 01-138 Warsaw, Poland; mfalis04@gmail.com (M.F.); parawan1970@wp.pl (P.R.-R.); an.lewandowska@igichp.edu.pl (A.L.); olimpiawitczak@tlen.pl (O.W.-J.); m.sobiecka@igichp.edu.pl (M.S.); monika.szturmowicz@gmail.com (M.S.); w.tomkowski@igichp.edu.pl (W.Z.T.); 2Department of Genetics and Clinical Immunology, National Research Institute of Tuberculosis and Lung Diseases, Płocka 26, 01-138 Warsaw, Poland; ulka100@gmail.com (U.L.); adriana.rozy@gmail.com (A.R.); 3Department of Radiology, National Research Institute of Tuberculosis and Lung Diseases, Płocka 26, 01-138 Warsaw, Poland; kasiabp67@gmail.com (K.B.); lilaja@wp.pl (L.J.); 4Department of Respiratory Physiopathology, National Research Institute of Tuberculosis and Lung Diseases, Płocka 26, 01-138 Warsaw, Poland; monika.franczuk@gmail.com; 5Outpatient Clinic, National Research Institute of Tuberculosis and Lung Diseases, Płocka 26, 01-138 Warsaw, Poland; zolnowskabeata@gmail.com; 62nd Department of Lung Diseases, National Research Institute of Tuberculosis and Lung Diseases, Płocka 26, 01-138 Warsaw, Poland; justynagryczka.pl@gmail.com

**Keywords:** MUC5B, IPF, treatment effect, survival

## Abstract

Idiopathic pulmonary fibrosis (IPF) is a rare disorder concerning elderly people, predominantly men, active or former smokers, with a progressive nature and leading to premature mortality. The cause of the disease is unknown. However, there are some risk factors, among which genetic predisposition plays a role. The aim of our single-centered observational study was to assess the correlation between single nucleotide polymorphism (SNP) of the *MUC5B* gene (rs35705950) and the disease course, antifibrotic treatment effect, and survival in patients with IPF. A total of 93 patients entered the study, of whom 88 were treated with either nintedanib or pirfenidone. The GG genotype was found in 28 (30.1%) subjects, while the GT or TT genotypes were found in the remaining 65 (63.4%) and 6 (6.5%) patients, respectively. The T allele minor allele frequency (MAF) accounted for 38.2% of the whole group. Patients with different genotypes did not differ significantly regarding age, sex, pulmonary function tests’ results, response to the antifibrotic treatment, or survival. However, we found a survival advantage in female patients and patients with higher pre-treatment TL,co. Treatment with antifibrotics significantly decreased the magnitude of FVC and TL,co decline compared to the time before treatment initiation, regardless of *MUC5B* status. In conclusion, we found high prevalence of T allele of *MUC5B* gene in patients with IPF; however, it showed no influence on disease trajectory, survival, or antifibrotic treatment effect in the presented cohort.

## 1. Introduction

Idiopathic pulmonary fibrosis (IPF) is one of the best-recognized and most prevalent interstitial lung diseases (ILDs). The annual incidence of the disease in Europe was recently estimated as 0.9–1.3/100,000 inhabitants, and the prevalence as 3.3–2.5/100,000 [1]; however, the exact numbers are difficult to predict as the diagnosis is complex, and patients are often misdiagnosed [2]. The median survival after diagnosis in IPF patients was reported as 3– 5 years before the implementation of the antifibrotic drugs and increased to 5–7 years in patients who received antifibrotic treatment [3,4,5]. Early diagnosis is crucial for timely treatment initiation, but it is difficult due to insidious and nonspecific symptoms [2,6]. Nevertheless, there are some known risk factors for developing the disease, like advanced age, male sex, cigarette smoking, and genetic predisposition [6,7].

The genetic signature of IPF has been studied for years [8,9,10,11,12]. However, no single nucleotide polymorphisms (SNPs) or pathogenic variants responsible for disease development have been found until the present day. The most prevalent genetic abnormalities involved genes related to telomere length (*TERT*, *TERC*), surfactant production (*SFPTA1*, *SFPTA2*, *SFPTC*), mucus turnover (*MUCB*), and others [13]. Genetic disorders are particularly important in familiar forms of pulmonary fibrosis (FPF). Nevertheless, genetic testing is not widely available. According to a recent Italian study, only 15% of ILD specialists have access to the molecular laboratory performing tests for FPF [14]. 

The *MUC5B* gene is located on chromosome 11p15.5 and encodes the high molecular weight glycoprotein mucin 5B, which increases the viscosity of mucus gel in the airways [15]. Mucin 5B is a key molecule in maintaining lung immune homeostasis and mucociliary clearance, and in relation to this, it plays an important role in controlling infections [15]. The lack of MUC5B in mice resulted in impaired airway clearance and decreased survival due to disseminated infections. MUC5B also influences macrophage accumulation and activity, with increase of apoptosis, if absent [15]. On the other hand, the increase in MUC5B concentration in distal airspaces, as a result of gain-of-function rs35705950 polymorphism, increased the intensity of fibrotic response to bleomycin in mice [16].

Multiple studies were performed to assess the presence of MUC5B-positive cells in the lungs of IPF patients. Nakano et al. confirmed the significance of rs35705950 of *MUC5B* gene promoter region in increasing the expression of *MUC5B* in bronchiolar epithelial cells in IPF lungs [17]. The role of SNP in the promoter region of the *MUC5B* gene (rs35705950) as a risk factor for IPF is well-established [18]. It is estimated that *MUC5B* polymorphism (the presence of T minor allele) accounts for 30–35% risk of developing lung fibrosis and is the strongest single risk factor for developing IPF [19,20]. It may be used as a marker of preclinical pulmonary fibrosis and the progression of radiological features of fibrosis [21,22]. With this marker, it may be possible to identify the cohort of subjects at increased risk of IPF development. However, patients with the presence of a minor allele (T) are thought to have a less severe disease trajectory [23]. The mechanism of this effect is not clear. It is thought that increased mucus production may enhance host defense mechanisms and reduce the number of infections, which may worsen the disease course. Additionally, patients with *MUC5B* rs35705950 polymorphism tended to present UIP pattern in HRCT less frequently, favoring atypical radiological patterns [24]. The role of *MUC5B* polymorphism in the improved response to treatment with interferon—gamma1b was also taken into consideration, suggesting a similar relationship to the other treatment modalities [23]. However, only one study on a relatively small group of patients has been published discussing the relationship between the *MUC5B* polymorphism and the response to antifibrotic treatment [25]. Previously, we analyzed the cohort of patients with hypersensitivity pneumonitis and found that the presence of the T allele predicted a faster decline in FVC irrespectively to immunomodulatory treatment [26].

Therefore, we aimed to investigate the group of IPF patients treated with antifibrotic drugs (either pirfenidone or nintedanib) to determine if the presence of minor allele of the *MUC5B* gene influenced the treatment outcome and survival.

## 2. Results

A total of 93 patients (70 males and 23 females) were included in the study. The characteristics of the study group are presented in Table 1. The mean age at diagnosis was 68.9 ± 8 years, and the mean follow-up duration was 2102 ± 982.5 days (70.1 ± 32.7 months). Seventy-eight (83.9%) patients were former or current smokers, with a median of 29 (IQR: 25–45) packyears of smoking. Almost all patients had mild or moderate disease at diagnosis: Gender–Age–Physiology (GAP) index stage I—67 subjects (72%); stage II—25 subjects (26.9%); stage III—1 person (1.1%)

Forty-seven (50.5%) patients were prevalent cases, and forty-six (49.5%) were incident cases. Incident cases were defined as those diagnosed within one year before their inclusion in the trial. The detailed comparison between the incident and prevalent cases is presented in Appendix A.

Incident cases were older at diagnosis as prevalent ones (mean age 71.65 ± 8.2 vs. 66.42 ± 7.1 years, respectively, *p* = 0.002, unpaired *t*-test). The observational period was significantly shorter in incident cases than in prevalent cases (47.9 ± 18.7 vs. 91.7 ± 28.9 months, respectively, *p* < 0.0001), and the death rate was significantly lower among incident cases (34% vs. 63%, *p* = 0.0036). Prevalent cases had exhibited UIP pattern in HRCT more often than incident ones (89% vs. 71%, *p* = 0.036).

The referenced GG genotype was found in 28 (30.1%) subjects and GT or TT genotypes in the remaining 65 (69.9%) patients. The minor allele (T) frequency (MAF) was 38.2% in the whole group. Male and female subjects did not differ significantly according to the genotype distribution and MAF (*p* = 0.7984).

The prevalence of the GG vs. GT/TT genotype was equally distributed among incident and prevalent cases. The GG genotype was present in 23.9% of incident cases and 36.2% of prevalent cases, whereas the GT or TT genotype was found in 76.1% of incident cases and 63.8% of prevalent cases (*p* = 0.1976, chi-square test). MAF was 41.3% in incident cases and 35.1% in prevalent cases (*p* = 0.45, Fisher exact test).

Eighty-eight (94.6%) patients from the study group were treated with antifibrotic drugs. Of those without treatment, one person was referred to another center, and the functional parameters were not available; one did not consent to the treatment; one was disqualified from the treatment because of severe comorbidities, including advanced lung cancer; finally, two were lost upon follow-up. Two of those patients had GG genotype, one TT and two GT (MAF 40%).

Among those who were treated, 43 (48.9%) patients received pirfenidone only, 29 (32.9%) received nintedanib only, and 16 (18.2%) were subsequently treated with both antifibrotic drugs. No differences were noted regarding age, sex, baseline PFTs, and *MUC5B* status between the different treatment groups. All but one person were treated for at least 12 months and had PFTs performed after a year from treatment initiation.

We compared patients with GG and GT/TT genotypes and found no differences in demographic or functional variables between the two groups apart from higher desaturation during 6MWT in patients with the GT/TT genotype at diagnosis. Patients with the GG genotype suffered from lung cancer more often, but the difference was not significant (Table 1).

We analyzed pulmonary function trajectories (FVC %pred and TL,co %pred) in treated patients. The changes within one year before the treatment initiation and after 12 months of antifibrotic treatment were considered. We found a significant decrease in FVC %pred and TL,co %pred during the year preceding treatment initiation and stable parameters after 12 months of treatment (Figure 1A,B and Table 2). Only 8 patients (9.1%) experienced a decrease in FVC of 10% or more after one year of antifibrotic treatment; in 36 (40.9%) subjects, FVC increased, and in 43 (48.9%), FVC remained stable.

The patients with the GG genotype had the same disease trajectories as the patients with the GT/TT genotype. There were no differences in the magnitude of the decline in FVC %pred before and after treatment initiation and TL,co %pred before and after treatment initiation among those two groups of subjects (Table 3). Additionally, there were no differences in the FVC and TL,co decline among patients with UIP and probable UIP patterns in HRCT—Table 3. The example HRCT scans of patients with UIP and probable UIP pattern are presented in Appendix A.

During the observational time, 44 people died or had a lung transplant. The median survival in the presented group was high (7.69 years). There were no differences in survival among patients with GG and GT/GG genotypes—Figure 2A. Median survival in patients with the GG genotype was 7.69 years, and in the group with the GT/TT genotype—8.25 years (*p* = 0.94).

Patients in GAP stage 1 had significantly better survival than those in stage 2 or 3 (9.02 vs. 5.55 years, *p* = 0.0009)—Figure 2B. Women survived longer than men (10.1 vs. 6.5 years, *p* = 0.0153)—Figure 2C. Survival was longer in the group with probable UIP pattern in HRCT, but the difference did not reach statistical significance—Figure 2D. There were no differences in survival between patients diagnosed at an older or a younger age and between patients treated with different antifibrotics.

We also analyzed survival from treatment initiation in the same groups of patients to overcome immortal bias. We found significantly longer survival in patients with GAP stage 1 compared to those in stage 2 or 3 (5.71 vs. 3.42, *p* = 0262), and between men and women (4.42 vs. 6.19 years, *p* = 0.0326)—Table 4.

The characteristics of patients who died compared to those who survived are presented in Table 5. Survivors were more often women, had better baseline PFT results, and received antifibrotic treatment for a longer time. There were no differences in genotype nor in the magnitude of FVC and TL,co decline during the first year of AF treatment between deceased and survivors. Lung cancer was more prevalent in the group of non-survivors (22.7 vs. 6.1%, *p* = 0.0338).

In the Cox regression analysis, we found that male sex was the factor increasing the odds of death (OR 3.607, *p* = 0.0067), whereas higher baseline TL,co (%pred) had the opposite effect (OR 0.9643, *p* = 0.0236)—Table 6.

Of those who were treated, all patients were alive after the first year of treatment; in the second, third, and fourth year of treatment, 10 (11.4%), 10 (11.4%), and 11 (12.5%) patients died, respectively.

## 3. Discussion

The SNP (rs35705950) of the *MUC5B* gene, which results in increased minor (T) allele frequency (MAF), is a well-known risk factor for the development of both sporadic and familial forms of IPF [27,28]. We analyzed the cohort of 93 patients with IPF who were followed up in the single pulmonary department, most of whom were treated with antifibrotic drugs (pirfenidone or nintedanib). Half of them were incident cases, with the other half being prevalent. Surprisingly, the prevalent cases were younger at diagnosis than the incident ones. This difference may be related to the COVID pandemic, which caused significant delays in ILD diagnostics and difficulties in accessing lung disease centers during the years 2020 to 2022 [29].

On the other hand, the proportion of women was higher among incident cases, as well as the proportion of probable UIP pattern in HRCT. This may indicate better diagnostic accuracy in female subjects suspected of IPF and reflects the new Polish guidelines published in 2020, which allow diagnosing IPF without biopsy in patients with probable UIP in HRCT and proper clinical context. Therefore, the diagnosis of IPF could have been made in female patients older than 60 years, males older than 50 years who have probable UIP pattern in HRCT and traction bronchiectasis/bronchiolectasis in two (if moderate or severe) or four (if mild) lobes, or in patients older than 70 who have reticulation comprising more than 30% of lung volume. Additionally, if BAL was performed, high neutrophile count or low lymphocyte count in BALF also allowed diagnosing IPF in patients with probable UIP patterns in HRCT [30].

All subjects were tested for the presence of *MUC5B* genetic polymorphism. This was the first genetic study on IPF patients in Poland. The MAF in our cohort was 38.2%, the GT genotype was present in 63.4% of patients, and the TT genotype was present in 6.5% of patients. The prevalence of SNP in our cohort was similar to other published cohorts of IPF patients of European ancestry [10,27,31]. On the other hand, based on genome-wide association studies (GWAS), MAF in the general population of European ancestry was estimated to be about 10% [9]. In our cohort of hypersensitivity pneumonitis (HP) patients, we presented a MAF prevalence of 17%, which was higher than in the general population but lower than in presently analyzed patients with IPF [26]. Similar to other authors, our study showed no significant differences regarding age, sex, smoking status, and disease severity between IPF patients with GG and GT/TT genotypes [23,32,33]. The pulmonary function tests (i.e., FVC %pred., TL,co %pred., and 6-min walking test (6MWT) parameters) were also comparable in both groups apart from baseline (at diagnosis) desaturation during 6MWT, which was significantly higher in the group of patients with the presence of T allele. This finding seems to contradict the data presented by other authors, who suggest a milder disease in patients with GT or TT genotypes [23]. On the other hand, van der Vis et al. found such a relationship only in patients with familial pulmonary fibrosis but not in sporadic cases, in whom *MUC5B* had no influence on the disease course [33]. Similarly, Sterclova et al. also reported a lack of differences in disease trajectories related to the *MUC5B* polymorphism [34].

Out of all our patients, 95% received antifibrotic treatment. We analyzed the disease trajectories before and after treatment. Within 12 months before treatment, the FVC %pred and TL,co %pred. decreased significantly, whereas after a year of treatment, they remained stable. These findings support the data on antifibrotic treatment efficacy from other cohorts [35,36,37,38]. The *MUC5B* genotype did not influence the treatment effect in our cohort of patients. Our findings are in line with the only study that reflected a similar problem in a similar group of IPF patients treated with antifibrotics, published by Biondini et al. [25].

In our cohort, the *MUC5B* genotype did not influence survival. Conversely, Peljito et al. showed significantly better survival in untreated IPF patients with the GT and TT genotypes compared to the GG genotype. This observation was independent of patients’ sex, age, and baseline pulmonary function tests [23]. On the other hand, it is known from other studies that the *TOLLIP* gene polymorphism correlated with the N-acetylcysteine (NAC) treatment outcomes in patients with IPF [39].

However, we observed significant survival advantage in females, as well as in patients with GAP stage 1 compared to GAP stages 2 and 3. Additionally, we observed that among the survivors, more patients had probable UIP pattern in HRCT, but this difference did not reach statistical significance, most probably due to the small number of subjects. Other authors showed that IPF patients presented with probable UIP survived longer than those with UIP pattern [40,41]. On the other hand, in our study, lung cancer was more prevalent in non-survivors. The negative influence of lung cancer on the survival of patients with IPF has been reported earlier [42]. Nevertheless, in our cohort, in the Cox proportional hazards regression analysis, only the male sex increased the odds of death, whereas higher baseline TL,co had a protective effect. Median survival in our study group was almost 8 years from the diagnosis, significantly longer than reported in the literature before the antifibrotic era. Additionally, we noticed a decrease in FVC %pred. of 10% or more in the first year of treatment, which only accounted 9% of patients. We noticed 40% improved their FVC, and 48% had stable disease. These results show an overall positive effect of antifibrotic treatment (improvement or stabilization of FVC %pred.), regardless of *MUC5B* status.

Our study has limitations. First, this is a single-center study that includes not only incident but also prevalent cases, influencing the time to treatment initiation; the study group is relatively small as well. Nevertheless, the patients are well-characterized and diagnosed according to the recent international and national guidelines. Some other studies also reported comparable groups of patients with similar or contradictory results, which shows the need to perform a large multicenter study related to this topic—Table 7. On the other hand, such smaller studies may be used in meta-analyses that increase the reliability of presented outcomes.

Second, immortal bias may be present (some patients with more severe disease could have died before the study started to enroll). To overcome this problem, however, we assessed survival not only from the diagnosis of IPF but additionally from the beginning of treatment. Third, due to the pandemic period that occurred just after the study had begun, some planned assessments (e.g., 6MWT) could not be performed and the recruitment was less efficient.

## 4. Material and Methods

### 4.1. Study Group

The research group included 93 consecutive IPF patients admitted to the First Department of Lung Diseases between 1 March 2019 and 31 December 2022.

All patients gave written informed consent for the trial participation and publication of the results. The trial was accepted by the institutional regulatory board (Bioethical Committee of the National Research Institute of Tuberculosis and Lung Diseases in Warsaw, Poland), document number KB-19/2019, date of approval 27 February 2019.

All patients received the IPF diagnosis based on recent international and Polish guidelines [30,43].

The study endpoints were defined as death or lung transplant during the observational period. The follow-up was censored on 30 September 2024.

### 4.2. Genetic Testing

Genetic testing was performed on all patients. The presence of a single nucleotide polymorphism (SNP) in the promoter region of the *MUC5B* gene rs35705950 was tested with the usage of a commercially available TaqMan SNP Genotyping Assay (C_1582254_20). The detailed procedure description was published elsewhere [26]. Shortly, a blood sample was taken during venipuncture and processed to receive high-quality DNA material. After that, two allele-specific TaqMan minor groove binder (MGB) probes with a fluorescence quencher and a primer pair uniquely aligned with the studied genome region were used. Reactions were performed on 96-well plates with negative and positive controls on Applied Biosystems 7500 Fast Dx Real-Time PCR Instrument (Life Technologies Holdings Pte Ltd., Singapore). Minor allele frequency (MAF) was calculated by dividing the number of the minor allele (T) by the number of all alleles.

### 4.3. Pulmonary Function Tests

Pulmonary function tests (forced vital capacity, FVC, transfer factor of the lungs for carbon monoxide, TL,co, 6-min walking test, 6MWT) were performed according to the usual schedule for patients treated with antifibrotic drugs, i.e., every six months. The data regarding the pre-treatment period were extracted from the patients’ medical files. Master Screen Body/Diffusion (Jaeger, Wuppertal, Germany, 2002; CareFusion, Wurmlingen, Germany, 2017) was used routinely to perform pulmonary function tests, according to the ERS/ATS guidelines [44,45]. The measurements’ results were presented as absolute, and the percentage of predicted values, according to the Global Lung Function Initiative, referenced equations [46]. The transfer factor of the lungs for carbon monoxide (TL,co), a measure of gas exchange efficiency, was measured with a single-breath method using helium or methane (CH4) gas as the marker, as described previously [26,47].

### 4.4. Imaging

HRCT scans were performed locally with the use of GE Healthcare Revolution GSI scanner. Images were assessed by radiologists experienced in interstitial lung disease according to the international and Polish guidelines [30,43]. The assessments were not additionally re-evaluated for the purpose of this study.

### 4.5. Statistical Analysis

The statistical analysis was performed using GraphPad Prism 10.0.1 (170) on 25 July 2023 (GraphPad Software, LCC, San Diego, CA, USA). The values were presented as mean ± SD if there was a normal distribution, and presented as median and range if the distribution differed from normal. A between-group comparison in two groups was assessed using the T-Student test, Mann–Whitney test, or Wilcoxon test, where appropriate, for continuous variables. Chi-square test or Fisher exact test were used for categorical variables. Survival analysis was performed using Cox proportional hazards regression. Kaplan–Meier curves for the whole group and specified variables were drawn. Survival analysis was performed and Kaplan–Meier curves were drawn with median survival assessment. *p* values of < 0.05 were considered statistically significant.

## 5. Conclusions

Our study undertook the important problem of biomarkers predicting responses to antifibrotic treatment in patients with IPF. We hypothesized that the presence of *MUC5B* genetic polymorphism may serve as such a predictor. Only one single study addressed this issue before. However, we found no clear influence of *MUC5B* polymorphism on the disease trajectory in treated patients. Furthermore, there was no influence on survival. On the other hand, significant positive survival predictors were the female sex and higher baseline TL,co (%pred). Our data showed the need for further research on predictors of treatment response in the group of patients with IPF.

## Figures and Tables

**Figure 1 ijms-26-02218-f001:**
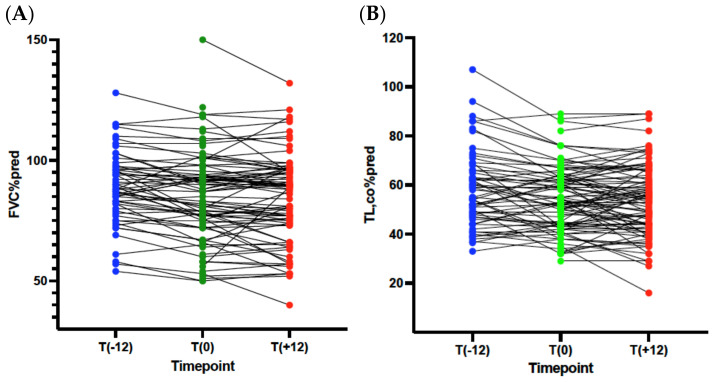
Changes in PFTs before and after treatment. (**A**). FVC %pred, (**B**). TL,co %pred. FVC %pred—forced vital capacity, %predicted; TL,co %pred—transfer factor of the lungs for carbon monoxide, %predicted; T (−12)—within a year before treatment initiation, T (0)—at treatment initiation, T (+12)—after 12 months of treatment.

**Figure 2 ijms-26-02218-f002:**
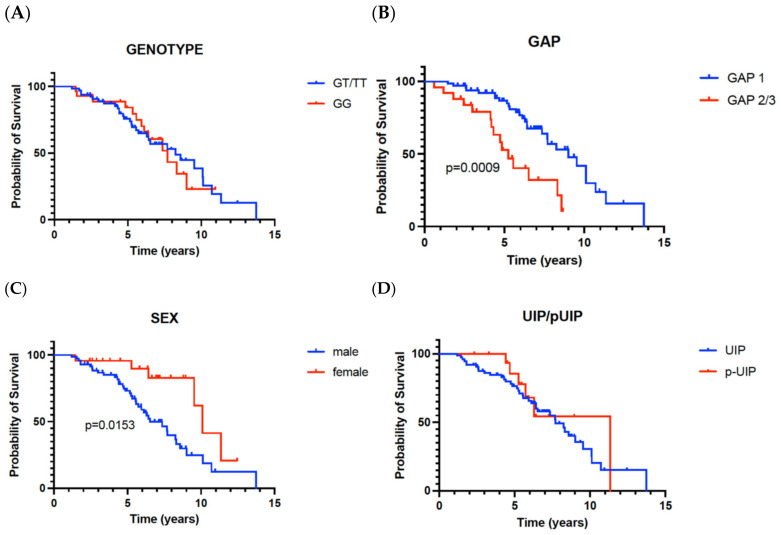
Kapplan–Meier survival curves according to different factors. (**A**)—genotype (GG versus GT/TT), (**B**)—GAP (stage 1 versus stages 2 or 3), (**C**)—sex (male versus female), (**D**)—HRCT pattern (UIP versus probable UIP). GAP—Gender–Age–Physiology Index, UIP—usual interstitial pneumonia, pUIP—probable usual interstitial pneumonia.

**Table 1 ijms-26-02218-t001:** Baseline characteristics of the study group and separately in the groups with different *MUC5B* genotypes.

	Whole Group (*n* = 93)	GG Genotype (*n* = 28)	GT/TT Genotype (*n* = 65)	*p* Value
Gender				
Males, *n* (%)	70 (75.3)	21 (75)	49 (75.4)	>0.99
Age at dgn, yrs (mean ± SD)	68.9 ± 8	67.18 ± 8.4	69.65 ± 7.8	0.1888
Smoking status, *n* (%)				
never	15 (16.1)	5 (17.9)	10 (15.9)	0.7676
former/active	78 (83.9)	23 (82.1)	54 (84.4)	
Packyears, median (IQR)	29 (15–45)	30 (20–45)	26 (15–45)	0.4529
Length of follow-up, mths (mean ± SD)	70.1 ± 32.7	70.4 ± 31.2	69.9 ± 33.6	0.9461
Baseline FVC, %pred (mean ± SD)	90 ± 18.4	90.3 ± 20.6	89.9 ± 17.6	0.9386
Baseline TL,co, %pred (mean ± SD)	58.8 ± 15.6	58.74 ± 14.4	58.83 ± 16.2	0.9798
Baseline 6MWD, m (mean ± SD)	480.5 ± 98	494 ± 102.2	475.2 ± 96.7	0.4513
Baseline desaturation, % (median, IQR)	4 (1–7)	1 (0–7.25)	4 (2.5–7)	0.0188
GAP index, *n* (%)				
I	67 (72)	22 (78.6)	45 (69.2)	0.4536
II + III	26 (28)	6 (21.4)	20 (30.8)	
HRCT (*n* = 92)				
UIP	74 (80.4)	24 (88.9)	50 (76.9)	0.2537
Probable UIP	18 (19.6)	3 (11.1)	15 (23.1)	
Antifibrotic treatment, *n* (%)	88 (94.6)			
Pirfenidone	43 (48.9)	10 (38.5)	33 (53.2)	0.4397
Nintedanib	29 (32.9)	10 (38.5)	19 (30.6)	
Both (consecutively)	16 (18.2)	6 (23)	10 (16.2)	
Time to treatment initiation, months (median, IQR)	17.8 (6.2–39.7)	23.9 (4.98–42.4)	16.4 (6.31–35.6)	0.5259
Length of treatment, m (mean ± SD)	45.5 ± 19.8	46.6 ± 20.2	45 ± 19.8	0.75
Death/transplant, *n* (%)	44 (47.3)	13 (46.4)	31 (47.7)	>0.99
Lung cancer, *n* (%)	13 (14)	7 (25)	6 (9.2)	0.0559

FVC—forced vital capacity, TL,co—transfer factor of the lungs for carbon monoxide, GAP—Gender–Age–Physiology Index, HRCT—high-resolution computed tomography, UIP—usual interstitial pneumonia, SD—standard deviation, IQR—interquartile range.

**Table 2 ijms-26-02218-t002:** Functional parameters before and after treatment.

	T (−12)	T (0)	T (+12)	*p* Value T (−12) vs. T (0)	*p* Value T (0) vs. T (+12)
FVC, %pred (mean ± SD)	88.8 ± 14.7	85.36 ± 15.6	84.1 ± 18.1	<0.0001	0.4441
TL,co, %pred (mean ± SD)	58.08 ± 15.6	54.7 ± 12.8	53.0 ± 14.0	0.0029	0.3055

T (−12)—timepoint within 12 months before treatment initiation, T (0)—timepoint at treatment initiation, T (+12)—timepoint after 12 months of treatment; FVC—forced vital capacity, TL,co—transfer factor of the lungs for carbon monoxide.

**Table 3 ijms-26-02218-t003:** Lung function trajectories in patients with different *MUC5B* genotypes and different HRCT patterns.

	Whole Group	GG Genotype	GT/TT Genotype	*p* Value	UIP	pUIP	*p* Value
FVC decline, 12 months before treatment initiation, %pred (median, IQR)	3 (0–7.5)	3 (1–7.75); *n* = 16	4 (−1.5–7); *n* = 45	0.6993	4 (0–8)	2 (−0.5–2.75)	0.6616
TL,co decline, 12 months before treatment initiation, %pred (median, IQR)	3 (−3–7)	2.5 (−2.25–6); *n* = 16	3 (−3–10.5); *n* = 45	0.8549	2 (−3–8)	3 (−4.75–7.25)	0.8485
FVC decline, 12 months after treatment initiation, %pred (median, IQR)	1 (−3–4)	2 (−3–5.5); *n* = 25	0 (−3–2.475); *n* = 62	0.1869	1 (−3–5)	−1 (−3.75–2)	0.2846
TL,co decline, 12 months after treatment initiation, %pred (median, IQR)	0 (−5–5)	−1 (−3.5–4.5); *n* = 25	2 (−5–5.25); *n* = 62	0.6831	0 (−5–5)	1 (−2.75–5.75)	0.7020

FVC—forced vital capacity, TL,co—transfer factor of the lungs for carbon monoxide, UIP—usual interstitial pneumonia, pUIP—probable usual interstitial pneumonia, IQR—interquartile range.

**Table 4 ijms-26-02218-t004:** Median survival from the diagnosis and from the beginning of treatment in different patient groups.

Group	Median Survival from the Diagnosis (Years)	*p* Value	Median Survival from the Beginning of Treatment (Years)	*p* Value
GG genotype	7.69	0.95	5.07	0.9295
GT/TT genotype	8.25		5.3	
UIP	7.7	0.3697	5.1	0.3201
Probable UIP	11.35		6.19	
Age 70 or older	8.25	0.8166	5.07	0.9432
Age less than 70	7.69		5.3	
GAP 1	9.02	0.0009	5.71	0.0262
GAP 2 or 3	5.55		3.425	
Male	6.52	0.0153	4.42	0.0326
Female	10.1		6.19	

UIP—usual interstitial pneumonia, GAP—Gender–Age–Physiology Index.

**Table 5 ijms-26-02218-t005:** The characteristics of deceased and survived patients.

Variable	Deceased, *n* = 44	Survived, *n* = 49	*p* Value
sex			
Male, *n* (%)	38 (54.3)	32 (45.7)	0.0292
Female, *n* (%)	6 (26.1)	17 (73.9)	
HRCT pattern			
UIP	38 (51.35)	36 (48.65)	0.1972
Probable UIP	6 (33.3)	12 (66.7)	
Genotype			
GG	13 (46.4)	15 (53.6)	>0.9999
GT/TT	31 (47.7)	34 (52.3)	
FVC (0) (%pred.), mean (±SD)	79.6 (±17.9)	90.7 (±17.25)	0.003
TL,co (0) (%pred.), mean (±SD)	49.9 (±13.5)	58.4 (±11.6)	0.0018
Lung cancer, *n* (%)	10 (22.7)	3 (6.1)	0.0338
FVC slope 0–12 (%), median (IQR)	0 (−3–5)	1 (−3.75–3.75)	0.4545
TL,co slope 0–12 (%), median (IQR)	0 (−3–8)	2 (−5–5)	0.6646
Time to treatment (days), median (IQR)	637 (203–1342)	429 (123.5–911)	0.1154
Treatment duration (days), median (IQR)	1140 (743–1613)	1475 (1006–1924)	0.0432

FVC—forced vital capacity, TL,co—transfer factor of the lungs for carbon monoxide, GAP—Gender–Age–Physiology Index, HRCT—high-resolution computed tomography, UIP—usual interstitial pneumonia, SD—standard deviation, IQR—interquartile range.

**Table 6 ijms-26-02218-t006:** Factors influencing survival in patients with IPF.

Variable	OR	95% CI	*p* Value
Male sex	3.607	1.524 to 10.02	0.0067
Age at diagnosis	1.029	0.9817 to 1.079	0.2407
UIP pattern in HRCT	1.084	0.3960 to 2.530	0.8621
FVC (0),%pred.	0.9967	0.9751 to 1.018	0.7678
TL,co (0), %pred.	0.9643	0.9338 to 0.9946	0.0236
Lung cancer	1.918	0.7500 to 4.309	0.1384

FVC—forced vital capacity, TL,co—transfer factor of the lungs for carbon monoxide, HRCT—high-resolution computed tomography, UIP—usual interstitial pneumonia, OR—odds ratio, CI—confidence interval.

**Table 7 ijms-26-02218-t007:** Review of the studies addressing the issue of the relationship between *MUC5B T* allele and disease course and survival in patients with IPF.

Ref. No	Author/Year	N° of IPF Patients	Treatment	MAF (%)	Influence on Disease Course (Yes/No)	Influence on Survival (Yes/No)	Independent Prognostic Factor (Yes/No)	Other Independent Prognostic Factors
23	Peljito/2013	580 (81 centers), validation cohort—148	No/IFN-γ1b	37–39	N/A	Yes-better	Yes	N/A
25	Biondini/2021	88	Pirf—51, Nint—37	42	No	Yes-better	No	FVC, Time to respiratory failure
32	Jiang/2015	187	N/A	19	Yes-worse	Yes-worse	N/A	N/A
33	Van der Vis/2016	115 (sporadic)	N/A	27	N/A	No	N/A	N/A
34	Sterclova/2021	50	“according to the guidelines”	N/A	No	N/A	N/A	N/A
Present study	Lewandowska/2025	88	Pirf—43, Nint—29, both (subsequently)—16	38	No	No	No	TLco, Female gender

Ref. N°—reference number, IPF—idiopathic pulmonary fibrosis, MAF—minor allele frequency, IFN-γ1b—interferon-γ1b, Pirf—pirfenidone, Nint—nintedanib, N/A—not applicable, FVC—forced vital capacity, TL,co—transfer factor of the lungs for carbon monoxide.

## Data Availability

Raw data are available from the corresponding author on request. A minimal dataset was uploaded as required.

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
