# Peer review of "MUC5B* Polymorphism in Patients with Idiopathic Pulmonary Fibrosis—Does It Really Matter?"

_ijms, 2025, doi:10.3390/ijms26052218_

Round 1

Reviewer 1 Report

Comments and Suggestions for Authors Pulmonary fibrosis is a progressive lung disease characterized by the thickening and stiffening of the lung interstitium due to excessive accumulation of extracellular matrix components, primarily collagen.   This pathological remodeling impairs gas exchange and leads to reduced pulmonary function and respiratory distress. The two treatment options, pirfenidone and nintedanib have been shown to slower disease progression in patients.    The submitted manuscript entitled „MUC5B Polymorphism in Patients With Idiopathic Pulmonary Fibrosis–Does It Really Matter? by K. Lewandowska and collogues analyzed the role of MUC5B polymorphism and their role treated and untreated fibrosis.    Finally, the there is no correlation of the GT or TT genotypes compared to wild GG genotype in terms of patients survival. However, the authors found differences in survival based of on sex and gender-age physiology index. The authors analyzed a cohort of finally 88 patients, which leads to relatively small subgroup. This may be the reasons why most comparisons were not significant. A more complex statistical analysis, with different covariates would also be helpful. Even its known, that MUC5B expression is important in pulmonary fibrosis, the authors addressed the important question if MUC5B polymorphism is important in the context of pharmacological treatment.   Additional comments:  1. The main question is the role of the polymorphism of MUC5B in patients with fibrosis. In addition survival depending on the treatment was analyzed.  2. The study addresses a valid question about the importance of this polymorphism. As there were no differences for MUC5B found, the authors compared other factors like sex and GAP. However, this was analyzed by other studies. therefore I wouldn't say, that the content of this manuscript really fill a gap in the field of fibrosis. 3. Finally only the information, that there is no role of the polymorphism in MUC5B in patients with fibrosis 4. What specific improvements should the authors consider regarding the methodology? From my point of view, the number of patients is too low to analyze the question. Especially for additional information of the different pharmacological treatment. As the power of the study is not very high, one have to be careful with the conclusion. With a larger cohort possible differences may be present. 5. References appropriate 6. Comments on the tables and figures. I found them clear, but as mentioned before no significant differences were found.   Over all. from the abstract the manuscript sounds interesting. After reading the parts several times, I was really left with the question what this study added to the field of fibrosis. An the answer is - very little!

Reviewer 2 Report

Comments and Suggestions for Authors

The research manuscript “MUC5B Polymorphism in Patients With Idiopathic Pulmonary Fibrosis–Does It Really Matter?” is a very well written clinical study involving 88 patients diagnosed with idiopathic pulmonary fibrosis. The authors make the point that the MUC5BT allele does not confer a milder or slower disease progression than the G allele. The manuscript is interesting to the scientific community because it adds to general knowledge. The references are well chosen and the results are comprehensively presented. However, there are some concerns with the manuscript which need to be addressed:

Major point:

1: The authors should comment on the power of their study. The number of people studied is very important for the conclusions reached when dealing with risk of developing disease or the progression of disease. The number of subjects included hare is 88, a very low number. Please comment on this in the discussion. How does this compare to other studies, where a connection was found between T allele and survival?

Minor comments:

1: Please do not call the GG genotype wild. It is the reference, nothing else.

2: Do not use contracted form (Line 33: didn’t differ significantly).

3: Please explain to laypeople what TL,co means.
